# Prevalence and Awareness of Medication Overuse Headache among Undergraduate Students at the University of Belgrade

**DOI:** 10.3390/brainsci14090938

**Published:** 2024-09-19

**Authors:** Aleksandra Radojičić, Ana Milićević, Mirjana Ždraljević, Marta Jeremić, Dajana Orlović, Milija Mijajlović

**Affiliations:** 1Faculty of Medicine, University of Belgrade, 11000 Belgrade, Serbia; milicevic.anci@gmail.com (A.M.); milijamijajlovic@yahoo.com (M.M.); 2Neurology Clinic, University Clinical Center of Serbia, 11000 Belgrade, Serbia; arsenijevicmirjana0905@gmail.com (M.Ž.); marta.jeremic@gmail.com (M.J.); dajanaorlovic10@gmail.com (D.O.)

**Keywords:** medication overuse headache, prevalence, awareness, undergraduate students, University of Belgrade

## Abstract

Background: Medication overuse headache (MOH) is a prevalent and potentially preventable secondary headache disorder linked to the excessive use of medications intended for primary headache management, particularly migraine. Aim: The aim of our study was to assess the prevalence of MOH among undergraduate students and explore their awareness. Methodology: This observational cross-sectional study included 401 active undergraduate students from the University of Belgrade. Data were collected through an anonymous online questionnaire which was distributed among student groups and via social media. The questionnaire specially designed for this study was developed in accordance with established guidelines for headache epidemiological research. Results: Among the surveyed students, 10 of them (2.5%) met the criteria for the diagnosis of MOH. Awareness of MOH was noted in 149 (37.2%) students, with higher awareness among medical students and those aged 22–25 years. Despite this awareness, there was no significant difference in MOH occurrence between those aware and unaware of the condition (aware 2.7% vs. unaware 2.4%, *p* = 1.000). Additionally, significant gaps in education and communication about MOH were evident. Limitations: Participants were recruited through convenience sampling from a single university at one time point. The questionnaire was not specifically validated in the student population, and the data relied on self-reporting. Conclusions: Our study highlighted a notable prevalence of MOH among undergraduate students, with a substantial portion exhibiting awareness of its risks. Despite this awareness, our findings suggest ongoing gaps in education and communication regarding MOH, emphasizing the need for targeted interventions.

## 1. Introduction

Over the past few decades, medication overuse headache (MOH) has garnered the attention of clinicians and researchers as one of the most prevalent yet still underdiagnosed types of secondary headaches. Population studies estimate that at least 2% of the world’s population use analgesics daily [1], and 4% suffer from chronic daily headaches [2]. MOH is considered to be caused by the regular overuse of medications intended to treat a pre-existing primary headache disorder [3]. In most cases, the pre-existing primary headache is migraine, while tension-type headaches (TTHs) are less common [4]. Medication overuse refers to the use of basic pain relievers (such as paracetamol and non-steroidal anti-inflammatory drugs—NSAIDs) for more than 15 days a month, or the reliance on opioids, triptans, and combinations of pain relievers for more than 10 days each month [3]. This condition affects approximately 59 million people worldwide, with an adult prevalence in the general population typically ranging from 1 to 2% and varying between 0.5% and 7.2% across different countries [5,6]. Besides geographic variations, there are also demographic differences. MOH affects women more often than men, with a ratio of 3.5 to 1, reflecting the higher prevalence of migraine among women, which is two to three times greater [7]. While it can occur in childhood, it typically peaks in the fourth decade of life and declines with advancing age [8].

MOH is a persistent and incapacitating headache condition that represents a major health, social, and economic challenge globally. Since 2015, the Global Burden of Disease study has not specifically addressed the burden of MOH, as it has been reassigned to migraine and TTH, both of which significantly contribute to the global burden, ranking third and fifteenth, respectively, among neurological disorders in terms of their impact on quality of life and daily functioning [9]. Identifying and managing MOH can be complex, often requiring the involvement of various medical specialties [10]. Furthermore, patients who initially respond successfully to treatment for MOH can experience relapse rates ranging from 20% to 60% in subsequent years [11,12,13]. Given its secondary nature, it is presumed that prevention is possible through the avoidance of frequent analgesic intake and raising awareness of the appropriate use of acute pain medications [14]. Educational strategies pointed towards migraine patients with frequent use of analgesics and triptans resulted in notable reductions in medication usage and fewer headache days and prevented the onset of MOH in a study by Fitch and colleagues [15]. The prevention of MOH involves recognizing the issue of medication overuse, which remains insufficiently addressed, not only within the general population but also by healthcare professionals [16]. Several headache and MOH awareness campaigns, including student and general public education, have been conducted with variable success [16,17,18,19].

While awareness and preventive strategies for MOH continue to be areas of uncertainty, they represent important targets for future intervention. To date, no studies have specifically examined MOH awareness in Serbia, and the prevalence of this burdensome disorder has only been assessed in the post-conflict regions of Kosovo and Metohija, in areas inhabited predominantly by ethnic Serbs [20]. In that study, the overall prevalence of MOH in the adult population was 2.9%, with rates ranging from 0.4% to 2.5% in individuals aged 18–35 years. Our study aims to address this gap by evaluating both the prevalence of MOH and the varying levels of awareness about the condition among students at the University of Belgrade.

## 2. Materials and Methods

This observational cross-sectional study was conducted at the University of Belgrade, Serbia, from 15 December 2023 to 15 January 2024. The inclusion criterion was that the participants were active undergraduate students at the University of Belgrade. The study excluded participants who declined to participate.

According to official data, the University of Belgrade has around 90,000 active students [21]. This number was used to calculate the minimum required sample size in this study using the OpenEpi (Version 3.01) website to maintain a 95% confidence interval and a 5% *p*-value [22]. The calculated value was 383. Students at the University of Belgrade are enrolled across 31 faculties and categorized into four scientific fields, social and humanistic sciences, medical sciences, natural and mathematical sciences, and technical and technological sciences, with over 6500 students enrolled in medical sciences programs [21]. Data were collected through an anonymous online questionnaire which was distributed among student groups and via social media. For the distribution of the questionnaire, representatives of the Students Parliament (a student representative body that facilitates communication between students and the university administration) from each of the four faculty groups were contacted and asked to disseminate the questionnaire to their peers. A link to the survey was distributed via student Viber and WhatsApp groups. Additionally, the survey was posted on the official Facebook page of University of Belgrade students, with respondents encouraged to forward the link to their fellow students. Participation was entirely voluntary, with all participants assured anonymity and confidentiality throughout the process.

All completed questionnaires were included for analysis.

The questionnaire specially designed for this study was developed in accordance with established guidelines for headache epidemiological research [23], with the HARDSHIP questionnaire serving as a guideline [24]. It contained 17 questions, organized in 4 domains, predominantly in the form of multiple-choice questions, supplemented by 3 checkbox questions and a single open-ended question (Appendix A). The screening and diagnostic questions for primary headaches and MOH, as well as healthcare-related questions, had previously been translated into Serbian through the “Lifting the Burden” translation protocol and validated for a prior study assessing the prevalence of primary headaches in adults within a post-conflict region of Serbia [20]. For the purposes of our study, these questions were minimally adapted to suit the online format. Before full deployment, the online questionnaire was piloted among a small group of 20 undergraduate students to identify any potential issues with its clarity and understandability.

The first part of the questionnaire collected data on demographic characteristics, such as gender, age, and whether the participant was studying at the Faculty of Medicine. Participants from other faculties were not further categorized. The second part of the questionnaire gathered data on the type, reason for use, and knowledge of the side effects of acute pain medications. The third part focused on detecting primary headaches. Participants were asked if they had experienced headaches in the past 12 months that were unrelated to cold, hangover, or head injury. Those who responded affirmatively were subsequently asked about the frequency of their headaches, pain quality, localization, associated symptoms, intensity, duration of untreated or unsuccessfully treated headaches, and the frequency of acute pain medication consumption. The final question educated them about the possibility of developing MOH as a consequence of regular analgesic use. The assessment of awareness about MOH was based on a multiple-choice question evaluating knowledge of the adverse effects of analgesic use. It was assumed that individuals who identified chronic headaches as a side effect of acute pain medications were aware of MOH. For the diagnosis of primary headaches, the ICHD-3 diagnostic criteria were utilized [3]. A case of migraine was defined in a subject experiencing headaches which last 4–72 h, who had at least 2 of the following characteristics: (1) unilateral location, (2) pulsating quality, (3) moderate or severe pain intensity, (4) aggravation by or causing avoidance of routine physical activity, and nausea and/or vomiting or photophobia and phonophobia. A case of TTH was defined in a subject with a headache which lasts from 30 min to 7 days, who had at least 2 of the following characteristics: (1) bilateral location, (2) pressing or tightening (non-pulsating) quality, (3) mild or moderate intensity, (4) not aggravated by routine physical activity and no nausea or vomiting, and no more than one of photophobia or phonophobia.

A case of MOH was defined in a subject experiencing headaches on ≥15 days per month and reporting the intake of NSAIDs or paracetamol on ≥15 days per month, or combination analgesics, triptans, or ergotamines on ≥10 days per month in the last 3 months. Primary headaches and MOH were diagnosed independently by two neurologists (AR and MŽ) based on the questionnaire responses.

Statistical analysis was conducted using IBM SPSS Statistics 17 (IBM, Armonk, New York, NY, USA). Frequencies were reported as absolute numbers and percentages, with intergroup comparisons of frequencies analyzed using Fisher’s exact test. A significance level of *p* < 0.05 was applied to determine statistical significance.

The study activities received ethical approval from the Institutional Review Board of the Medical Faculty University of Belgrade, Serbia (Approval No 500/05). All participants provided informed consent before participating, and the study ensured the confidentiality and anonymity of participants’ data. Ethical standards aligned with national and institutional policies for human subjects’ research were strictly followed.

## 3. Results

This study included 401 participants, out of which 294 (73.3%) were female. The majority of participants, 207 (51.6%), were between 22 and 25 years old. The second most common group consisted of participants younger than 22 years (136 or 33.9%), while only 57 (14.2%) were in a category of 25 years or older. Out of the total sample, 229 students (57.1%) were enrolled in the Faculty of Medicine, while 172 (42.9%) were from other faculties. 

Three hundred fifty subjects (87.3%) reported taking acute pain medications in the past 3 months. Among them, 218 participants (62.3%) cited headache as the primary reason for their use, with others mentioning different types of pain. Fifty-one students (12.7%) indicated that they did not use these types of medications.

Among the participants, the most frequently used analgesics were non-steroidal anti-inflammatory drugs (NSAIDs), utilized by 312 individuals (77.8%). Triptans and ergotamines, medications used to treat migraine attacks, were used the least, with only five participants (1.2%) taking them. The profiles of the medications and the frequency of their consumption in the past 3 months are presented in Table 1. 

Awareness of MOH as a side effect of regular analgesic use was noted in 149 (37.2%) individuals. 

Of the total sample, 289 students (72.1%) reported having headaches unrelated to cold, hangover, or head injury in the last year, while 59 students (14.7%) consulted a doctor due to headaches. Regarding the type of primary headache, 227 participants (56.6%) reported characteristics consistent with tension-type headache, whereas 61 participants (15.2%) referred to headache characteristics that met the criteria for migraine. Regarding headache frequency, 194 students (48.4%) experienced infrequent headaches (once a month or less), while 14 students (3.5%) reported having headaches 15 or more days per month. Ten participants (2.5%) met the criteria for MOH diagnosis.

Headache as a reason for taking acute pain medications was more frequently reported in the group of participants diagnosed with MOH compared to those without MOH (*p* = 0.014). Additionally, migraines were significantly more common among subjects with MOH compared to those without MOH diagnosis (*p* = 0.01). Individuals experiencing MOH were more inclined to seek medical help for symptom management (*p* = 0.001) (Table 2).

A higher percentage of respondents from the Faculty of Medicine recognized chronic headache as a potential adverse effect or regular intake of medications for acute pain treatment (*p* = 0.001). The awareness of MOH was significantly higher among students aged 22–25 years (*p* = 0.001). Furthermore, there was a notable difference in the profile of commonly used analgesics between participants with different levels of awareness of MOH. Paracetamol (*p* = 0.005), combined analgesics (*p* = 0.044), and triptans/ergotamines (*p* = 0.021) were more frequently used among those aware of MOH compared to their unaware counterparts, while NSAIDs showed a similar frequency of consumption in both groups (Table 3).

In response to the question “If you were aware that regular use of painkillers could lead to chronic headache, what changes would you make?”, 49 participants (12.2%) indicated they would not change anything, 232 (57.9%) stated they would reduce their usage, and 118 (29.4%) said they would seek medical advice. In the group of patients with MOH, seven of them answered that they would reduce the use of analgesics, one answered that they would seek help, and two that they would not change anything.

## 4. Discussion

Our study analyzed the types and reasons for pain-relieving medication use, awareness of their link to chronic headaches, and the prevalence of MOH among undergraduate students at the University of Belgrade. The majority of participants used analgesics in the 3 months prior to the survey, primarily for headache treatment. In a study conducted at the University of Birmingham, headaches were also the leading reason for analgesic use in student populations (85%) [18]. The most commonly used medications in descending order were paracetamol, NSAIDs, and combination analgesics, while migraine-specific therapy such as triptans and ergotamines were used by only 1% of individuals, although one out of seven participants reported primary headaches meeting the ICHD-3 diagnostic criteria for migraine. A multicenter study conducted in several European countries and Australia also indicated that paracetamol and NSAIDs are the most commonly used analgesics, with their consumption steadily increasing [1]. We identified a smaller group of study subjects, 12.7%, that did not use any medication for acute pain treatment, while comparative results yielded proportions of 29.5% [25]. In total, 72.1% of our study participants suffered from primary headaches. TTH followed by migraine was the most common type, while MOH was present in 2.5%. The results obtained in our specific study group consisting of students still correspond to population-based studies. The prevalence of primary headaches, including migraines and tension-type headaches, was as expected and consistent with data from the Danish population study [26], and a previous study examining primary headache prevalence among adults in Serbia [20]. The frequency of MOH in our study also aligns with the data from studies that included children and adolescents, showing a range from 0.3% in Taiwan to 3.3% in Italy, with migraine being the most common underlying headache disorder [27,28,29]. We found that the majority of individuals suffering from MOH are female, as previously demonstrated in numerous studies investigating patients with chronic headaches. Similar results have been obtained in the general population, where women had 50% more MOH than men [30].

After investigating the distribution and demographics of the study participants regarding their awareness of MOH and its causes, our results indicate a well-developed awareness of MOH among students at the University of Belgrade, particularly among future doctors. Medical students also showed a higher response rate to the questionnaire than their colleagues from other study programs, which may reflect their greater interest in health-related issues and the direct relevance of the topic to their education. The overall level of awareness was over 37%. Similar, albeit slightly lower, proportions were illustrated in a study of Lai J. et al., which showed that 38% of students in the healthcare-educated group recognized MOH and its causes, compared to 14% of non-healthcare-educated individuals [18]. The dependence of awareness level by age was illustrated in our study, where we found that the majority of subjects aware of MOH were aged 22–25 years. Similar results were also reported in the literature, with a mean age of 37.4 years; however, most studies included a broader age range of participants, encompassing both students and the general adult population, which may explain the difference in the mean age [31]. Increasing awareness of MOH before its diagnosis could play a crucial role in the treatment outcomes of patients with headaches. In light of the importance of heightened awareness and primary prevention, various educational programs have already been implemented. The impact of prevention of MOH was demonstrated by Cerlsen L. et al., showing that by implementing nationwide prevention programs, a 7% increase in MOH awareness in the general population is possible [16]. Moreover, several randomized trials investigating patient education’s impact on managing MOH in Europe have also demonstrated modest benefits [32,33]. It is common for headache patients to self-manage their pain with over-the-counter (OTC) medications. Unfortunately, discussions about MOH between patients and healthcare providers are often insufficient [34,35]. The individuals with MOH in our study were more likely to seek professional medical opinion for their headaches. However, it is evident that the level of awareness about MOH did not differ significantly between those with MOH and those without the condition. This suggests that while MOH may prompt individuals to seek medical help, awareness about the condition itself may not be well understood or effectively communicated, highlighting a potential gap in patient education and healthcare provider awareness. Additionally, increased awareness of MOH in a group of our participants was not associated with lower MOH occurrence. This could imply that other factors, such individual susceptibility, coping mechanisms, comorbidities, or treatment adherence, may play significant roles in the development and persistence of MOH, beyond just awareness alone. Understanding these factors could be crucial for developing effective strategies to manage and prevent MOH in clinical practice.

The findings from our study underscore the need for targeted public health interventions and clinical practice changes to increase awareness and reduce the prevalence and impact of MOH in Serbia. From a public health perspective, educational campaigns should be implemented to inform students and the general population about the risks associated with the frequent use of over-the-counter analgesics while promoting behavioral changes. This could involve workshops, informational campaigns, and partnerships with student organizations across universities in Serbia. This is supported by the results of our analysis, where 70% of students with MOH reported that they would have reduced the use of analgesics if they had known that they could cause chronic headaches. Subsequent evaluations of these programs should be conducted to assess their effectiveness in raising awareness about MOH. Additionally, further studies should explore the long-term outcomes of such educational interventions on MOH prevention.

From a clinical perspective, our findings emphasize the importance of educating both the public and healthcare professionals, particularly those in student health services, who should be trained to recognize patterns of medication overuse and to provide early interventions. We also propose screening for headache disorders and medication overuse in routine health assessments for students. This can help identify individuals at risk of developing MOH and offer alternative pharmacological and non-pharmacological strategies to support the discontinuation of self-medication and improve headache management.

There are several limitations in our study. Participants were recruited through convenience sampling, which may not represent the broader population. Those who chose to participate could have different perspectives than those who did not, potentially affecting the generalizability of our results. Since participants were students recruited from a single university the findings may not apply universally across different demographics or regions. Furthermore, the questionnaire was adapted from a previously validated tool for headache research in adults and minimally adjusted for an online format, but it was not specifically validated in the student population. While the questionnaire was piloted on a small group of students to identify potential issues, it may not have captured the full spectrum of headache or medication use behaviors unique to younger adults. Additionally, the data relied on self-reporting, which is subject to recall bias. Finally, being cross-sectional, our study captured data at a single point in time, preventing any conclusions about causality between awareness and MOH prevalence. Further research employing a more representative sampling strategy, ensuring population-specific validation of the questionnaire and using longitudinal design, may be necessary to explore the underlying mechanisms and potential interventions to prevent MOH.

## 5. Conclusions

MOH remains a significant health challenge globally, arising from the frequent use of analgesics intended for primary headache disorders, primarily migraine. Our study highlighted a notable prevalence of MOH among undergraduate students, with a substantial portion exhibiting awareness of its risks. Despite this awareness, our findings suggest ongoing gaps in education and communication regarding MOH, emphasizing the need for targeted educational interventions that address both awareness and behavioral changes in medication use to mitigate the burden of MOH and improve patient outcomes. Future research should explore longitudinal approaches to better understand the dynamics of MOH development, particularly in younger populations such as university students, where early educational interventions may exert a lasting impact on MOH prevention.

## Figures and Tables

**Table 1 brainsci-14-00938-t001:** Frequency of painkiller use in the past three months.

Frequency	Number (%)
Rarely (average once a month or less)	192 (47.9)
2–9 days per month	142 (35.4)
10–14 days per month	7 (1.7)
15 days or more per month	4 (1)
I haven’t used painkillers	56 (14)

**Table 2 brainsci-14-00938-t002:** Comparative analysis of demographics, medication use, and headaches in participants with and without MOH.

Variable	With MOH N = 10	Without MOH N = 391	*p*-Value
Female gender	10 (100%)	284 (73.6%)	0.07
Age			
≤21	3 (30%)	133 (34.1%)	0.817
22–25	2 (50%)	202 (51.8%)	
≥25	5 (20%)	55 (14.1%)	
Faculty Type			
Medical Faculty	4 (1.7%)	225 (98.3%)	0.260
Other Faculties	6 (3.5%)	164 (96.5%)	
Painkillers			
Does not use	0 (0%)	51 (13%)	0.642
Paracetamol	4 (40%)	164 (41.9%)	1.000
NSAIDs	9 (90%)	303 (77.5%)	0.712
Combined analgesics	4 (40%)	165 (42.2%)	1.000
Triptans/ergotamines	0 (0%)	5 (1.3%)	1.000
Reasons for using painkillers			
Headache	10 (100%)	208 (53.3%)	0.014 *
Other pain	0 (0%)	130 (33.3%)	
Knowledge of adverse effects			
Loss of appetite	3 (30%)	62 (15.9%)	0.231
Stomach pain	8 (80%)	190 (48.6%)	0.083
Nausea, vomiting, diarrhea	7 (70%)	216 (55.2%)	0.535
Chronic headache	4 (40%)	145 (37.1%)	1.000
Fatigue	4 (40%)	111 (28.4%)	0.495
Skin reactions	4 (40%)	156 (39.9%)	1.000
Liver damage	6 (60%)	207 (52.9%)	0.762
Kidney damage	5 (50%)	164 (41.9%)	0.755
Unfamiliar with adverse effects	2 (20%)	105 (26.9%)	1.000
Headache not related to cold, hangover, or head injury	10 (100%)	279 (71.4%)	0.068
Consulting a doctor due to headache	6 (60%)	53 (13.6%)	0.001 *
Type of primary headache			
Tension-type	5 (50%)	222 (56.9%)	0.752
Migraine	5 (50%)	56 (14.4%)	0.010 *

NSAID = nonsteroidal anti-inflammatory drugs; * statistically significant.

**Table 3 brainsci-14-00938-t003:** Comparative analysis of demographics, medication use, and headaches in participants aware and unaware of MOH.

Variable	Aware of MOH N = 149	Unaware of MOH N = 252	*p*-Value
Female gender	108 (73%)	185 (74.9%)	0.794
Age			
≤21	37 (25%)	99 (39.4%)	0.001 *
22–25	96 (64.9%)	110 (43.8%)	
≥25	15 (10.1%)	42 (16.7%)	
Faculty Type			
Medical Faculty	123 (82.6%)	105 (42.2%)	0.001 *
Other Faculties	26 (15.3%)	144 (84.7%)	
Painkillers			
Paracetamol	77 (51.7%)	90 (35.9%)	0.005 *
NSAIDs	116 (77.9%)	195 (77.7%)	0.829
Combined analgesics	74 (49.7%)	95 (37.8%)	0.044 *
Triptans/ergotamines	5 (3.4%)	0 (0%)	0.021 *
Reasons for using painkillers			
Headache	88 (59.5%)	129 (51.4%)	0.439
Other pain	44 (29.7%)	86 (34.3%)	
Headache not related to cold, hangover, or head injury	110 (73.8%)	179 (71.3%)	0.277
Consulting a doctor due to headache	24 (16.1%)	35 (13.9%)	0.627
Type of primary headache			
Tension-type	85 (57%)	142 (56.8%)	0.697
Migraine	25 (16.8%)	36 (14.4%)	0.632
MOH	4 (2.7%)	6 (2.4%)	1.000

NSAID = nonsteroidal anti-inflammatory drugs; MOH = medication overuse headache; * statistically significant.

## Data Availability

The data presented in this study are available only upon request from the corresponding author due to privacy and confidentiality concerns regarding the participants’ information. Access to the data will be provided in accordance with the ethical guidelines and institutional policies governing the protection of personal data and research confidentiality.

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
