# Peer review of "Prevalence and Awareness of Medication Overuse Headache among Undergraduate Students at the University of Belgrade"

_brainsci, 2024, doi:10.3390/brainsci14090938_

Round 1

Reviewer 1 Report

Comments and Suggestions for Authors

Dear authors,

The topic is very interesting, and the manuscript is well designed. Here are some comments on your manuscript:

Abstract:  Mention of the sample size and key statistical findings. Ensure that the abstract reflects the study limitations.

Introduction: Clarify any specific gaps in the current literature that this study aims to fill, emphasising the novelty of your research.

Methodology: Provide more detail on the questionnaire used, including a brief description of its development and validation. Discuss whether the questionnaire was pre-tested or piloted among a small group of students to identify any potential issues before full deployment. Is there any bias due to the sampling method? Can you add questions to the supplement materials ? What types of questions are used ? multiple-choice, Likert scale...?

Explain the recruitment strategy in more detail. For example, describe how student groups were selected and how social media was utilised.

Clearly define any inclusion and exclusion criteria used to select participants, ensuring the sample is representative of the target population.

Line 81: How did you diagnose a headache? 

Line 90: Mention the specific ethical guidelines followed and provide the approval number or reference.

Line 99: Specify other faculties.

Line 136: after ? you missed "

Table 2.  I don't understand the variable. Medical faculty ? The table is not easy to understand; separate or highlight Pinkilers, Reasons for using pain Killers, etc.

Suggestions to improve Table 2 and Table 3.

Discussion: Expand the implications of the findings for public health and clinical practice. Suggest interventions or educational programmes based on the study's findings.

Conclusion: Reinforce the key message of the study. What is the importance of awareness and education in preventing MOH? Explain.

Provide more concrete recommendations for future research directions.

Reference: Please check once more; you missed one citation. Ensure that all references are up-to-date. 

Author Response

Abstract:  Mention of the sample size and key statistical findings. Ensure that the abstract reflects the study limitations.

Thank you for this suggestion. In the abstract we have added your suggestions.

Introduction: Clarify any specific gaps in the current literature that this study aims to fill, emphasising the novelty of your research.

Thank you for this valuable suggestion. In the introduction we have added a paragraph explaining the gaps in the literature that we have been focused on filling with our study.

Methodology: Provide more detail on the questionnaire used, including a brief description of its development and validation. Discuss whether the questionnaire was pre-tested or piloted among a small group of students to identify any potential issues before full deployment. Is there any bias due to the sampling method? Can you add questions to the supplement materials ? What types of questions are used ? multiple-choice, Likert scale...?

We have added additional information as requested about the structure of the questionnaire, its development and validation. We have also added some instances of bias by sampling during the data collection while using the questionnaire in the Discussion section/study limitations. The questionnaire in English is provided as supplementary material.

Explain the recruitment strategy in more detail. For example, describe how student groups were selected and how social media was utilised.

We have explained in more details how students were recruited and how social media was used.

Clearly define any inclusion and exclusion criteria used to select participants, ensuring the sample is representative of the target population.

We have added a paragraph specifically assessing the inclusion and exclusion criteria.

Line 81: How did you diagnose a headache? 

We have added the segment describing diagnostic approach in details.

Line 90: Mention the specific ethical guidelines followed and provide the approval number or reference.

We have included a segment referencing ethical standards as well as the approval number received for this study by our ethical committee.

Line 99: Specify other faculties.

We have only collected data on whether the participant was studying at the Faculty of Medicine, while participants from other faculties were not further categorized.  We have clarified that in the revised manuscript.

Line 136: after ? you missed "

Thank you, “ was added in the indicated sentence.

Table 2.  I don't understand the variable. Medical faculty ? The table is not easy to understand; separate or highlight Pinkilers, Reasons for using pain Killers, etc.

Different sections representing different domains were highlighted. We added category Faculty type and data for Medical Faculty and Other Faculties.

Suggestions to improve Table 2 and Table 3.

Thank you, Tables 2 and 3 were improved

Discussion: Expand the implications of the findings for public health and clinical practice. Suggest interventions or educational programmes based on the study's findings.

We have added a paragraph with implications of our findings for both public health and clinical practice, along with suggestions for conducting further educational programs.

Conclusion: Reinforce the key message of the study. What is the importance of awareness and education in preventing MOH? Explain.

Thank you, we have added more explanation of our conclusion.

Provide more concrete recommendations for future research directions.

Such have been indicated in the added paragraph recommending educational programs and their evaluation. More specific suggestions for the design of further studies that could overcome the limitations of current study have also been added.

Reference: Please check once more; you missed one citation. Ensure that all references are up-to-date. 

                Thank you for your careful review and for pointing out the missing citation. We have corrected the mistake in the revised version.

Reviewer 2 Report

Comments and Suggestions for Authors

The authors present a cross-sectional online survey among students and ask for prevalence of headache disorders, knowledge about MOH and compare the results of medical students with students of other faculties.
However, some points have to be discussed:

Introduction:

The authors revere to the GBD study 2015, meanwhile there are publications on the 2021 study, therefore the authors should check if they can use more recent references (for example: GBD 2021 Nervous System Disorders Collaborators. Global, regional, and national burden of disorders affecting the nervous system, 1990-2021: a systematic analysis for the Global Burden of Disease Study 2021. Lancet Neurol 2024 Apr;23(4):344-381. doi: 10.1016/S1474-4422(24)00038-3. Epub 2024 Mar 14.

The most recent public awareness project about MOH was from the Denmark and should be included (for example: Carlsen LN et al. National awareness campaign to prevent medication-overuse headache in Denmark. Cephalalgia 2018 Jun;38(7):1316-1325. doi: 10.1177/0333102417736898. Epub 2017 Oct 10.)

Material and Methods:

The reviewer suggests to add the questionnaire in English as supplementary material. Diagnosing headache disorders using online questionnaires need the use of a validated questionnaire. The authors were asked to comment on the questionnaire in more details.

Results
Not surprising medical students showed a higher response rate to the questionnaire, they are more interested in health care topics. However, for the interpretation of the numbers more details on the overall student population at the university of Belgrade are needed. (How many students overall are the population for this study, how many from the medical faculty?)

Discussion

Please check references, obviously there are some errors [22,21,15], references and authors or topics mentioned in the manuscript does not fit to these references.

Author Response

Introduction:

The authors revere to the GBD study 2015, meanwhile there are publications on the 2021 study, therefore the authors should check if they can use more recent references (for example: GBD 2021 Nervous System Disorders Collaborators. Global, regional, and national burden of disorders affecting the nervous system, 1990-2021: a systematic analysis for the Global Burden of Disease Study 2021. Lancet Neurol 2024 Apr;23(4):344-381. doi: 10.1016/S1474-4422(24)00038-3. Epub 2024 Mar 14.

Thank you, the suggested reference is added to the revised introduction. We have also kept GBD study from 2015, since it was the last iteration to include MOH as a separate entity.

The most recent public awareness project about MOH was from the Denmark and should be included (for example: Carlsen LN et al. National awareness campaign to prevent medication-overuse headache in Denmark. Cephalalgia 2018 Jun;38(7):1316-1325. doi: 10.1177/0333102417736898. Epub 2017 Oct 10.)

The suggested reference has been included and cited in the Introduction, while the study results have been commented in the Discussion section (reference 16 in both, the original and revised manuscript).

Material and Methods:

The reviewer suggests to add the questionnaire in English as supplementary material. Diagnosing headache disorders using online questionnaires need the use of a validated questionnaire. The authors were asked to comment on the questionnaire in more details.

We have commented in the materials and methods section regarding the composition process of the questionnaire, its validation and structure. We provide the English version of the questionnaire as a supplement to the revised manuscript.

Results
Not surprising medical students showed a higher response rate to the questionnaire, they are more interested in health care topics. However, for the interpretation of the numbers more details on the overall student population at the university of Belgrade are needed. (How many students overall are the population for this study, how many from the medical faculty?).

Thank you for this observation, we have added more information on the student population numbers in the Materials and method section, and also included your comment in the Discussion.

Discussion

Please check references, obviously there are some errors [22,21,15], references and authors or topics mentioned in the manuscript does not fit to these references.

We have corrected the missing citation in the revised version of the manuscript and ensured consistency throughout the reference list.

Round 2

Reviewer 2 Report

Comments and Suggestions for Authors

Thanks for revising the manuscript according to the reviewers suggestions.